# Transient Gene Expression is an Effective Experimental Tool for the Research into the Fine Mechanisms of Plant Gene Function: Advantages, Limitations, and Solutions

**DOI:** 10.3390/plants9091187

**Published:** 2020-09-11

**Authors:** Alexander A. Tyurin, Alexandra V. Suhorukova, Ksenia V. Kabardaeva, Irina V. Goldenkova-Pavlova

**Affiliations:** Timiryazev Institute of Plant Physiology, Russian Academy of Sciences (IPP RAS), Moscow 127276, Russia; alexjofar@gmail.com (A.A.T.); sualsha@yandex.ru (A.V.S.); kabardaewa@yandex.ru (K.V.K.)

**Keywords:** transient transformation, plant systems, gene function discovery, location of gene product, protein–protein interaction, gene constructs

## Abstract

A large data array on plant gene expression accumulated thanks to comparative omic studies directs the efforts of researchers to the specific or fine effects of the target gene functions and, as a consequence, elaboration of relatively simple and concurrently effective approaches allowing for the insight into the physiological role of gene products. Numerous studies have convincingly demonstrated the efficacy of transient expression strategy for characterization of the plant gene functions. The review goals are (i) to consider the advantages and limitations of different plant systems and methods of transient expression used to find out the role of gene products; (ii) to summarize the current data on the use of the transient expression approaches for the insight into fine mechanisms underlying the gene function; and (iii) to outline the accomplishments in efficient transient expression of plant genes. In general, the review discusses the main and critical steps in each of the methods of transient gene expression in plants; areas of their application; main results obtained using plant objects; their contribution to our knowledge about the fine mechanisms of the plant gene functions underlying plant growth and development; and clarification of the mechanisms regulating complex metabolic pathways.

## 1. Introduction

Since the earliest stages in the research into the mechanisms underlying implementation of genetic information, researchers have clearly understood the importance of the gene expression control in eukaryotic cells, plants included. The first steps in the study of this problem were made owing to a deep insight into the mechanisms of transcription control as the first stage of gene expression and the easiest one for experiments in the methodical aspect. High-throughput technologies for transcriptome analysis, such as RNA-Seq and microarrays, have considerably clarified the key mechanisms underlying the regulation of gene transcription, expanded our knowledge about the transcriptional control of gene(s), and provided convincing experimental data demonstrating dynamic changes in transcriptomes of different plant species during their growth, development, and response to manifold environmental factors [1,2]. A large volume of data on gene expression obtained in comparative omic studies and, as a rule, meticulously processed with the help of available bioinformatics tools has made it possible to reveal the common transcriptome networks controlling a certain function and to predict the role of particular genes in cell processes [1,2]. It should be emphasized that the approaches of in silico analysis mainly rely on the predictive functional classification of genes constructed based on the search for and comparative analysis of the known homologous genes, as a rule, orthologs [1,3,4]. However, all this is insufficient to clarify the function of a particular gene as well as the fine mechanism of its operation. For example, a full understanding of the physiological role of a gene involved in a certain biological process requires in vivo genetic, molecular, and physiological–biochemical analyses. Before the advent of genome-wide sequencing, the functions of genes were analyzed using a “direct” genetics approach, namely, by comparison of the phenotypic or physiological effects of the natural or induced mutations in the sequence of a target gene [5]. Recently, the availability of genomic and transcriptomic data together with the designed efficient technologies of genetic modification of almost any plant species allows the gene function to be studied using the so-called “inverse” genetics: from the gene to the phenotype. This approach implies that the function of a gene in plants is studied either by varying its expression followed by analysis of the corresponding phenotype or by complementing the mutation, as a rule, using the models of T-DNA of induced mutants. The expression of a target gene can be varied using two main approaches: (i) by a complete or a partial “switch-off” of its expression, for example, using T-DNA mutations or RNA interference (RNAi) or (ii) via overexpression by producing transgenic plant lines that constitutively express the target gene at a very high level [5]. According to experimental data, such variations in gene expression can give abnormal phenotypes allowing for identification of more specific or finer effects in the function of the target gene [3,4,5].

The use of plant genetic engineering technologies, i.e., construction of transgenic plants, has made it possible to get unique experimental models of transgenic plants the manifold molecular and physiological–biochemical analysis of which gave unprecedented experimental confirmations for the functional roles of numerous plant genes [3,4,5]. Nevertheless, the main limitations of using the technology for obtaining transgenic plants are: (i) dependence of the transformation efficiency on the plant genotype; and (ii) considerable material and time expenditures when constructing plant transgenic lines, including selection of true transgenic individuals among a tremendous number of primary transformants, confirmation of integration of the transferred sequences and of the events of integration of a single copy into the plant genome, determination of the expression level for both mRNA and protein product, and so on (Figure 1I). Considering these limitations in the construction of stable plant transformants, and first and foremost, in terms of time expenditures [3,5], the demand for additional approaches to the experimental confirmation of the physiological role of plant genes has become apparent.

1—Plant growth; 2′—Sterilization of plant material; 2—Transformation/agroinfiltration/transfection; 3—Plant regeneration; 4—Selection of transgenic plants and molecular analysis; 5—Evaluation of the localization of a gene product in a plant cell or physiological role of the gene product in plant growth and development or role of the gene product in defense mechanisms when exposed to environmental factors.

A large pool of experimental data on agrobacterial transformation of different plant species clarified the key mechanisms of transfer, integration, and expression of heterologous genes in plants. According to experimental data, the process of transfer of a gene construct (which as a rule is represented by T-DNA) can be divided into two main stages: the entry of agrobacteria into plant tissue (transfection) and the integration of T-DNA into plant genome (transformation). Note that the first stage (transfection) is much more efficient than the second stage. Indeed, the second stage, integration, is frequently so inefficient that most rare transformed cells of the total plant cell population are selectable with the help of the genes coding for selective markers. Differential efficiency of these two stages means that millions of plant cells during a short time are transfected by T-DNA but not obligatory transformed, which allows only for a short-term (transient) expression of gene constructs in a plant cell [3,4,5]. These results formed the background for designing the approach of transient gene expression in plants.

According to the current opinion, the technology of transient gene expression in plant systems has considerable advantages over the stable expression, namely, (i) it does not require regeneration of the transformed cell; (ii) has no effect on the stability of host genome; and (iii) is independent of the position effects of the T-DNA integration sites. In addition, the use of this technology radically accelerates the study allowing the function of a target gene to be analyzed in 2 to 10 days after a gene construct is delivered to plant cells (Figure 1II,III). These advantages have made the transient expression system a powerful tool in the in vivo study of gene function by physiological and molecular characterization of diverse plant material [3,5]. A meta-analysis of research papers on transient expression of heterologous genes in plants has convincingly demonstrated that this technology is a trustworthy and already widespread methodical approach to clarification of the gene functions in plants (Appendix A).

This review elucidates the state-of-the-art experimental approaches to the research into the physiological role of plant genes using the strategy of transient expression with discussion of their advantages and limitations. Currently, it would be unjust to speak about insufficient number of publications describing similar basic protocols of transient expression for different plant species as well as the technical aspects associated with the effective use of this approach in assessing the functional role of the gene(s) in individual plant species [3,4,5,6,7] (Appendix A). Note also that numerous reviews describe the use of transient gene expression in plant systems to produce recombinant proteins, first and foremost, for medical purposes [5,8,9,10], assessing the activity of regulatory sequences (promoters, translational enhancers) [11], and other important aspects of plant molecular biology, such as the plant protein activity, the target gene expression regulation, the protein–protein interactions, and etc. [12,13,14,15,16]. However, no review in the available scientific literature so far considers and discusses the key components of transient expression strategy for different plant systems in the context of studying the physiological role of plant genes. Our goal in this review is (i) to consider the advantages and limitations of different plant systems and different methods of transient expression used to find out the role of target gene products; (ii) to summarize the current data on the use of transient expression approaches for the insight into fine mechanisms underlying the function of target gene products; and (iii) to outline the accomplishments in effective transient expression of target sequences in plants, including the use of viral suppressors of gene silencing, selection of agrobacterial strains, and plant growth conditions. Thus, the review will discuss the main and critical steps in each of the methods of this general strategy; areas of their application; the main results obtained using plant objects; their contribution to our knowledge about the fine mechanisms of the functioning of the plant genes involved in plant growth, development, and the response to environmental factors; and clarification of the mechanisms regulating complex metabolic pathways.

## 2. Transient Expression of Heterologous Genes: Plant Systems, Gene Constructs, and Their Delivery to Plant Cells

The plant systems used for transient expression include protoplasts [17,18], calluses, suspension cultures of plant cells [8], intact plants [4,5], as well as isolated plant organs [3] or specialized plant tissues [18,19].

Two types of gene constructs are mainly used for transient expression: (i) the expression vectors constructed involving binary Ti plasmids with gene cassettes as T-DNA or (ii) the vectors carrying modified genomic sequences of plant viruses, including those cloned into T-DNA cassettes [5,9].

Several approaches have been designed and are used for delivering the gene constructs for transient expression: transfection of protoplasts using polyethylene glycol (PEG) or electroporation [20], biolistics [21], and agroinfiltration [7]. Numerous studies have convincingly demonstrated their efficacy for characterization of gene functions, including assessment of subcellular localization of protein products of the target genes, study of the signal transduction pathways, and protein–protein interaction [22,23], as well as for clarifying the functional properties of regulatory sequences in plants and so on [1,24,25]. Each system of transient gene expression in plants as well as the approaches to the delivery of gene constructs along with its specific advantages has its own limitations and bottlenecks depending on the goal of a particular research.

Production of the protoplasts, deprived of the cell wall, is an effective approach to transformation of complexly organized plant tissues into the object more available for observation [26,27,28,29,30], which has made them quite popular in the studies into subcellular localization of the protein products of target genes as well as in testing of various regulatory sequences, such as promoters and translational enhancers [1,25]. Protoplasts also can be extremely useful when studying cell-autonomous regulatory processes and responses in quantitative and high-throughput ways. For instance, the protoplast also is a good model for ‘gene-for-gene’ research in plant immunity [12]. Notwithstanding, protoplasts have limited applicability to the cases requiring tissue and/or organismic context [3,31]. Note here that the transfection of protoplasts works well only for some plant species and requires complex preparatory procedures for their efficient transfection [32].

Biolistics can be used for different plant species. The delivery method has also proved useful for calluses or specialized plant tissues, for example, for the soybean aleurone layer or plant pollen tube [21,33]. The soybean aleurone layer is a biological proxy for the cotyledon of soybean since the single cell layer of the aleurone is comparable to the cotyledon tissue according to the morphological examination and comparative analysis of proteins and metabolomes. As a consequence, such plant system is a potentially useful platform for transient gene expression in the functional analysis of the genes that influence the seed characteristics of the soybean [33]. An additional example is a plant pollen tube that is also a good model for protein localization, trafficking and plant cell growth [21]. However, biolistics has certain limitations, including a low efficiency in the delivery of gene constructs, a relatively complex procedure for preparation of biological material, for example, ripe fleshy fruits, which are anatomically unsuitable for bombardment, as well as the need for specialized laboratory equipment and materials [32,34].

The suspension cultures of plant cells have been mainly obtained for the model plants, such as *Arabidopsis thaliana* or *Nicotiana tabacum* (BY-2). Their limitations are similar to those of protoplasts. For example, the BY-2 cells are undifferentiated cells; correspondingly, this tissue is suitable for testing only constitutive regulatory elements. In addition, the BY-2 cells lack certain specific metabolic pathways and biochemical components, for example, characteristic of seeds, which can vary depending of the plant species in both biochemical pathways and qualitative composition and distribution of storage compounds. All this interferes with the assessment of whether the gene constructs will work correctly in the seeds of the corresponding species as compared with the studied plant species [33].

The transient expression mediated by Agrobacterium, commonly known as agroinfiltration or agroinjection, utilizes the penetration of the solution containing agrobacteria into the intercellular space of plant tissues with the help of a needleless syringe or through vacuum [9]. According to the current opinion, it is technically easier to perform transient gene expression in plants using the agroinfiltration technology, which has several advantages. First, agroinfiltration makes it possible to overcome some critical stages associated with generation of protoplasts and their transfection: (i) when isolating protoplasts from an intact plant, the conditions of its growth are of importance; (ii) sterilization of plant tissue is necessary in some cases; and (iii) a strict control of the quality and quantity of protoplasts and plasmid DNA is necessary for transfection efficiency [32,35,36]. Second, both the transformation procedures and stress impact on the plant cell are minimized [37] (Figure 1II). Third, agroinfiltration provides a high transformation efficiency since the intercellular space represents up to one-third of the plant tissue volume and agrobacteria are actively delivered to the intercellular space. This provides an efficient access of agrobacteria to the majority of plants cells, thereby making the T-DNA transfer highly effective [10]. Additional advantages of this technology are high scalability of the process, allowing a large number of plants to be processed, and the possibility to perform multiple transient expression assays with several transgenic constructs on a single leaf [10,38]. Moreover, such experiment was conducted under native conditions with preservation of the tissue organization and cell integrity of the plant [37].

A necessary condition of a successful agroinfiltration of a plant is its susceptibility to *A. tumefaciens* infection and, among the susceptible plants, the compliance of different species with agroinfiltration, which depends on the genetic background of the host plant and considerably varies depending on the structural differences in the cuticle and the compactness of mesophyll cells [39]. *N. benthamiana* and the related Nicotiana species are the most popular host plants for agroinfiltration. However, improvement of the transient expression technology has allowed agroinfiltration to be used in many plant species (currently, over 40 species; Appendix A), including tree species. In addition to the leaf tissue, the petals of the tobacco, petunia, several dendrobium species, as well as tomato and strawberry fruits have been also successfully agroinfiltrated with transgenic constructs [10].

Nonetheless, the use of agroinfiltration for observation of individual plant cells in vivo is not a simple task. This is determined by the fact that the plant tissue cells in the majority of cases have rather complex configuration [23,32,34] and these specific features of the plant tissue structure and organization impose limitations on the use of agroinfiltration technology applied in the research into some aspects of the physiological role of genes, in particular, the studies of subcellular localization of target protein products [23,32,34]. Thus, the use of agroinfiltration encounters the problems of fine visualization of the target proteins in the plant cell compartments because of complex outlines of plant epidermal cells [31]. However, these limitations can be bypassed, for example, by using agroinfiltration followed by generation of protoplasts of the agroinfiltrated leaf regions, as has been demonstrated using *N. benthamiana* plants. Since the protocols for protoplast generation as well as agroinfiltration have been developed or improved for many model and nonmodel plants, this approach is also applicable to other plant species for which either (or both) approach(es) has been efficiently applied or either approach can be supplemented [31,40].

## 3. Main Research Areas Utilizing Transient Gene Expression in Plants

### 3.1. Localization of the Protein Products in Plant Cells

According to the current opinion, the function of a protein is tightly associated with its localization in the cell. In particular, the intracellular localization of enzymes gives key information for understanding of complex metabolic pathways. The general strategy in the studies assessing the localization of a target protein, as a rule, includes (i) in silico analysis of the amino acid sequence of the protein product of a target gene for predicting its localization; (ii) construction of the hybrid genes so that the target gene is transcriptionally and translationally fused with the sequence of a reporter gene, as a rule, the genes coding for fluorescent proteins; and (iii) determination of the localization of the protein product of the hybrid gene in one of the above described plant systems.

Currently, researchers have at their disposal numerous resources for in silico prediction of protein localization, for example, SignalP, ChloroP, iPSORT, TargetP, and MultiLoc [41]. The results of predictions give information not only about putative localization of the protein product, but also, and frequently, about the likely location of signal peptides (in the N- or C-terminal region), which can mediate the specific localization of the target protein. These predictions are used to design the gene constructs for experimental verification of the protein localization in the predicted compartments taking into account the information about localization of the signal peptide in the protein sequence (in the N- or C-terminal region) [38,42,43]. The gene constructs for experimental verification carry the hybrid gene where the target gene is transcriptionally and translationally fused with the gene coding for, as a rule, a fluorescent protein under the control of a strong constitutive promoter (Figure 2 part 1 or 2). Note that the selection of a particular fluorescent protein for localizing the target gene product is dictated by the predicted localization in the plant cell. In particular, green fluorescent protein has proved to be an effective reporter in many experiments on verification of target protein localization in the nucleus [44,45], cytoplasm [31,46], plasma membrane [46], Golgi apparatus [41], endoplasmic reticulum (ER) [31,41], tonoplast [47], mitochondria [48], and chloroplasts [41], while, in addition to green fluorescent protein, yellow fluorescent protein (YFP) and mCherry could be used to assess localization in peroxisomes, as recently demonstrated [38,43].

Note that control experiments are extremely important for a reliable localization of the protein products in plant cells and discarding of the false results. As a rule, these control experiments must include (i) positive control, which consists in colocalization of the target sequence and the specific markers of plant cell compartments (Figure 2 part 3) and/or (ii) negative control, i.e., a gene construct carrying only the gene coding for a reporter fluorescent protein under the control of the same promoter (Figure 2 part 4). According to experimental data, the proteins the subcellular localization of which in the particular compartment has been confirmed by reliable methods (for example, histochemical and immunohistochemical assays) can be used as specific markers for the corresponding plant cell compartments. Currently, the markers of this type are known and have been tested for localization in different cell compartments, including ER [41], peroxisomes [38,43], and chloroplasts (with the help of chloroplast autofluorescence) [41]. Note that the approach relying on co-infiltration with the Agrobacterium suspensions carrying different constructs with the target sequence and the sequence of organelle markers fused with different fluorescent proteins is, probably, the most effective for these purposes. This approach makes it possible to analyze the localization of several differently labeled fluorescent proteins in the same cells [41]. In our view, the constructs carrying reporter genes of fluorescent proteins fused with the known sequences providing specific localization also can be proposed as specific markers for different plant cell compartments, for example, the signal peptide of the Rubisco small subunit gene (for localization to chloroplasts) or the signal peptide of the pea (*Vicia faba*) B4 legumin gene together with SKDREL sequence in the 3′-terminal region of the reporter gene, coding for the amino acids from the C-terminal region of the protein (for localization to ER) [5,31].

Numerous examples confirm that transient expression of fluorescently labeled proteins is a simple and universal method for localizing the proteins of interest in the cell. This approach has given reliable data on the nuclear localizations of various transcription factors [44,49] and new localizations of the proteins in plant cell compartments, for example, oxidases with different substrate specificities [38,41], transferases [42,43], and many other proteins (Appendix A). These data suggest that several pathways involve many organelles and that metabolites migrate between different compartments [41]. Thus, the approaches involving transient expression have shown their efficiency in the assessment of localization of gene products in plant cells.

### 3.2. Study of the Physiological Role of Gene Products Involved in Plant Growth and Development

Two main approaches are as a rule used for clarifying the functions of the gene products when using the transient expression strategy: overexpression and silencing of the target gene either independently or in combination (Figure 3). Thus, the approaches utilized in transient expression are similar to the approaches used in the case of stable plant transformation. However, according to experimental data, these two approaches in transient expression strategy are, as a rule, effective for clarification of the functions of gene products the variation in expression of which leads either (i) to visible change in the phenotype of the used plant system or (ii) to the variations in protein activities in the corresponding metabolic pathways, which influences the qualitative and/or quantitative composition of the plant metabolome. On the other hand, the advantages of the transient expression of a target gene product are evident. First and foremost, this is the applicability of the method not only to the leaf tissue, but also to individual plant organs, in particular, fruits and berries, which considerably reduces the time and material expenditures as compared with the generation of stable plant transformants.

For example, recent functional analysis of the important genes involved in the biosynthesis of anthocyanins in strawberry fruits utilizing transient expression clarified the flavonoids pathway and enabled more accurate assessment of the contribution of each gene to the synthesis of the key compounds [3]. In particular, silencing of the gene coding for anthocyanidin-3-O-glycosyltransferase (*FaGT1*), involved in anthocyanin biosynthesis, allowed its role in the regulation of anthocyanin pigments to be identified (in particular, of the level of epiafzelechin, the substrate for anthocyanidin reductase, FaANR); this demonstrated the competition between two proteins, FaGT1 and FaANR, for the common substrate of anthocyanidin [3]. Another example of the study utilizing concurrent downregulation and overexpression of the genes coding for chalcone synthase (*CHS*) or eugenol synthase (*EGS*) during transient expression demonstrated the deviation of the flavonoid pathway to the synthesis of phenylpropene (eugenol), confirming the role of EGS in the contribution to production of volatile compounds responsible for the strawberry aroma [3]. A comparative study of expression of three tonoplast sugar transporters (*CmTST1*, *CmTST2*, and *CmTST3*) from melon plants has shown that overexpression of one of these genes, *CmTST2*, in the strawberry and cucumber fruits used as a model for transient expression increases accumulation of sucrose, fructose, and glucose in these fruits. Note that the *CmTST2* overexpression in strawberry fruits delayed their reddening but increased sugar accumulation at the final stage of fruit development. These results demonstrate that *CmTST2* most likely plays an important role in sugar accumulation in the melon fruits as well [47]. A transient overexpression in peach fruits of the gene coding for UDP-glucosyltransferase (*PpUGT85A2*) cloned from the same plant has convincingly demonstrated that the protein product of this gene controls glycosylation of a volatile compound, the monoterpene linalool, by catalyzing formation of its nonvolatile glycosylated species, linanyl-β-d-glucoside, which influences the taste of the fruit [43]. These examples demonstrate the capabilities of transient expression in the studies into physiological role of the genes involved in complex processes of plant (and their individual organs) growth and development.

### 3.3. Study of the Physiological Role of Gene Products Involved in Plant Responses to Biotic Environmental Factors

The strategy of transient expression also has been used as an effective experimental tool to gain insight into the function of the genes involved in the plant response to biotic environmental factors as well as into the intricate defense signaling pathways in different plant species, including the plants of the generation of stable transformants for which this is very time-consuming. For example, one of the cell wall proteins of the grapevine, polygalacturonase, was comprehensively studied using transient expression, and the silencing of this gene convincingly demonstrated its role in limiting the pathogen attack by the case study of *Botrytis cinerea*, the agent of gray mold [51]. The silencing of β-glucosidase in transient expression experiments has given convincing evidence that this gene is involved in the strawberry resistance to *B. cinerea*, which is most likely determined by an increase in both the PAL activity and content of phenolic compounds [52]. Recently, transient expression has been successfully used to clarify the role of a gene coding for a specific lectin. The unripe fruits with a downregulated expression of this gene displayed the signs of anthracnose, whereas the ripe strawberries with a lectin gene overexpression displayed a lower susceptibility to *Colletotrichum acutatum*. These experiments suggest that the gene in question plays a key role in the resistance of unripe strawberry fruits to this pathogen [53]. Transient expression of two genes coding for phosphatidylinositol transfer protein of the sugar cane in the *N. benthamiana* leaves increased the resistance of these model plants to the tobacco pathogens *Ralstonia solanacearum* and *Fusarium solani* var. *coeruleum*. Moreover, H_2_O_2_ accumulation and development of a hypersensitivity response to *R. solanacearum* and *F. solani* var. *coeruleum* inoculation were observed after a short-term overexpression in *N. benthamiana* as well as a change in the level of transcripts of the marker genes associated with the development of immunity in tobacco plants. Thus, it is convincingly demonstrated that one of the sugar cane genes (*ScSEC14p*) is tightly associated with the plant immunity and development of the hypersensitivity response, while another gene (*ScSEC14-1*) can be involved in some other mechanism of plant immunity [46]. Transient expression has also allowed for demonstration that an overexpression of the glyceraldehyde-3-P dehydrogenase genes of *Manihot esculenta* (*MeGAPCs*) rendered the plant less resistant to the bacterial blight. On the contrary, the cassava plants with lower *MeGAPC* gene expression caused by their transient silencing display an increased resistance to this pathogen. Moreover, a physical interaction of *MeGAPCs* with autophagy-associated protein 8b (MeATG8b) and MeATG8e and inhibition of autophagic activity were observable. Note that MeATG8b and MeATG8e downregulate the NAD-dependent MeGAPDH activities and contribute to the *MeGAPCs*-mediated disease resistance. Altogether, this study clarifies the important role of *MeGAPCs* in plant disease resistance via the interaction with MeATG8b and MeATG8e [54].

Thus, a change in the transient gene expression in plants (either overexpression or silencing) makes it possible to find out the degree to which the endogenous levels of a target protein contributes to the plant defense as well as to clarify the functions of the plant endogenous genes associated with pathogenesis or defense mechanisms [4].

It should be emphasized that the appropriate correct controls are extremely important when studying the functions of the genes involved in the plant response to plant pathogens and the defense signaling pathways in plants since the agroinfiltration itself can induce defense responses of the host, interfering with the interpretation of results [3,4]. The plants inoculated with the agrobacteria carrying an empty vector may be an adequate control [55].

### 3.4. Study of the Mechanisms of Regulation of Plant Metabolic Pathways

According to numerous experimental data, transient gene expression in plants has proved most effective for insight into the mechanisms underlying the regulation of complex metabolic pathways, first and foremost, of the secondary plant metabolites, which have a high potential as valuable products of plant origin for application in various areas. In particular, the role of key players in some metabolic pathways was demonstrated using transient expression as well as the possibility of their change aiming to purposefully increase the production of a target secondary metabolite. For example, transient expression has made it possible to find out that the specific MYB transcription factors are the key regulators of the flavonoid pathway and to identify their target genes in the metabolic pathways of the main flavonoid derivatives in the grapevine and grape berries, namely, flavonols, anthocyanins, and proanthocyanidins (PAs), which influence wine quality [56,57]. More particularly, the MYBA1 and MYBA2 transcription factors were experimentally confirmed to specifically regulate UDP-glucose:flavonoid 3-O-glucosyltransferase (UFGT) in the biosynthesis of anthocyanins [58] and MYBPA1 to be involved in the regulation of the biosynthesis of proanthocyanidins but not anthocyanin [59], while MYB5a and MYB5b to activate the promoters of the general flavonoid pathway genes [57]. In addition, the light-inducible MYBF1 transcription factor was demonstrated to specifically activate the expression of the gene coding for flavonol synthase as well as the genes involved in the synthesis of chalcones and flavanones before dihydroflavonols and other flavonoids [60]. A recent study has identified a new transcription factor, MYR R2R3, from the Chinese narcissus (*Narcissus tazetta* L. var. *chinensis Roem*) capable of repressing anthocyanin, flavonol, and proanthocyanidin pathways; it represses expression of the transcripts of the key genes coding for the enzymes involved in flavonoid biosynthesis. The results of this work have the potential for improving the coloration and structure of the flowers of this species [61]. Transient expression and silencing in the pear fruits and young leaves confirmed the biological function of the MYB12b transcription factor from the pear fruit. In particular, the overexpression of this transcription factor upregulates the flavonol biosynthesis, including the four main quercetin glycosides, via a positive regulation of the common gene of the flavonoid biosynthesis (chalcone synthase) and a gene of flavonol biosynthesis (designated PbFLS) [62]. The transient expression in *N. benthamiana* leaves of two splice isoforms of the WRINKLED1 (WRI1) transcription factor, belonging to the AP2/EREBP class of transcription factors of the pear fruits, upregulated expression of the target genes, such as *PKp-b1*, *ACP1*, and *KAS1*, and, as a consequence, increased the content of plant oil 4.3–4.9-fold as compared with the control. Thus, the role of this transcription factor as a key transcription regulator of the genes involved in the biosynthesis of fatty acids and oils in the leaf biomass was in general confirmed; as a consequence, this transcription factor has a potential for constructing the plants with a high content of biomass to increase the plant oil production [63]. Another recent paper reports the use of transient gene expression in plants to assess the utility of flexible linker sequences of different lengths in construction of the enzyme fusions—3,3′-β-carotene hydroxylase (CRTZ) and 4,4′-β-carotene oxygenase (CRTW), involved in the biosynthesis of a valuable ketocarotenoid, astaxanthin, a natural red dye with a high antioxidant activity. Transient expression of the hybrid genes in *N. benthamiana* showed that both genes transiently expressed as a fusion accumulated the astaxanthin levels similar to the transient expression of individual fragments. Note that the linker size had actually no effect on the enzyme activities. Altogether, the results favor a high potential of this approach for producing valuable compounds of a plant origin [64].

Recently, we used transient gene expression to examine the functional activity and substrate specificity of heterologous desaturases, in particular, to localize desaturases in different plant cell compartments (cytoplasm, chloroplast, and ER) in the case study of the *Synechococcus vulcanus* Δ9 acyl-lipid desaturase (*des*C). The chloroplast and ER localizations of this desaturase were provided by the leader sequences that guided the protein product of the target gene to the corresponding plant cell compartments, while the cytoplasmic localization was modeled by expression of the target gene without any leader sequences. The transient expression of Δ9 desaturase in the *N. benthamiana* and *N. excelsior* leaves caused a significant decrease in the content of palmitic acid (16:0) as well as an increase in the shares of oleic (18:1), linolenic (18:2), α-linolenic (18:3), and palmitolinolenic (16:3) fatty acids. Moreover, the utility of *N. benthamiana* and *N. excelsior* as model plants was convincingly demonstrated in the studies of functional activity and/or substrate specificity of various heterologous desaturases, not only Δ9 desaturase, but also the desaturases converting fatty acids 18:1 into polyunsaturated fatty acids, since the model plants (*N. benthamiana* and *N. excelsior*) tested in this study have significantly different fatty acid compositions and the C18/C16 ratio [55].

### 3.5. Study of Protein–Protein Interactions (PPI) and Their Subcellular Localization

Transient expression has also shown its efficacy in the in vivo studies of protein–protein interaction (PPIs) and subcellular localization. For these purposes, bimolecular fluorescence complementation (BiFC) analysis is most frequently used [65,66,67,68,69]. This approach makes it possible to qualitatively and quantitatively estimate and localize the PPIs in living cells. BiFC analysis assesses the interaction of two target proteins each of which is transcriptionally and translationally fused with one of the domains of a fluorescent protein via cofolding of the fluorescent protein into a β-barrel structure in the case of PPI of the target proteins, which restores the fluorophore, i.e., this approach is based on the irreversible self-assembly of two fluorescent protein fragments (Figure 4A). This PPI is visualizable with the help of a routine fluorescence or confocal laser scanning microscopy. Although the GFP cleavage assay in BiFC in general acted as an effective folding reporter, several pitfalls or limitations for its used in BiFC were initially noticed, including (i) poor detection in a real time mode and irreversible reconstitution of fluorescent chromophore; (ii) nonspecific interactions because of a high expression level of the fluorescent fragments, which resulted in a spontaneous self-assembly and accumulation of background signals; and (iii) insolubility or aggregation of large split fluorescent fragments, interfering with PPIs [65,66,69,70]. A recent improvement of this approach, which consists in tripartite complementation involving superfolder GFP (sfGFP) in plant cells in combination with the β-estradiol–induced expression cassette, allows these limitations to be overcome (Figure 4B). In particular, the β-estradiol–induced expression cassette makes it possible to control the induction of fusion proteins, which resolves the problem of cell toxicity caused by constitutive overexpression of this fusion; the effect of steric hindrances on PPIs in the tripartite system with split GFP is reduced to minimum because of the small size of the GFP fragments fused to the target proteins. The principle of the tripartite split GFP association consists in the following: each of the two GFP fragments, namely, β-strand 10 (designated by the authors as GF10; 20 amino acids) and β-strand 11 (GF11; 19 amino acids) are used for transcriptional and translational fusion with the studied partner proteins, while the detector fragment (designated GF1–9, 196 amino acids) is expressed as a separate expression cassette (Figure 4B). Once a PPI takes place, GF10 and GF11 are tethered and spontaneously associate with the detector fragment (GF1–9), which restores the full-length GFP and fluorescence [69] (Figure 4B). This experimentally confirms the efficacy of this approach as a potentially effective tool for assessing the interaction between membrane proteins in plants.

Thus, the approaches utilizing transient gene expression in plants are most informative for revealing of the fine mechanisms underlying the function of target gene products, namely, for estimation of the functions of the genes involved in the plant growth and development as well as in the response to the biotic environmental factors and the clarification of the regulation of complex metabolic pathways. Note that both heterologous (a target gene of one plant species is assessed in another plant species) and homologous (the target gene and the plant system are of the same origin) systems can be used for assessment of the functions of target gene products with the help of transient expression. When planning the corresponding experiments, the plant system and the method for delivery of gene constructs must be grounded, which is to a greater degree determined by the specific features of their applicability to studying different aspects of the gene function assessment and the nature of the target gene.

## 4. Specific Features of Vector Constructs, Agrobacterial Strains, and Nuances of Conditions for Effective Expression of Gene Products in Plants

Gene constructs (vectors) for ectopic expression of a gene or its silencing used for assessing the protein localization in the cell, PPIs, and enzyme activity are an important tool for characterization of the target gene product [3,5,24]. The experimental studies utilizing transient expression mainly use the available commercial vectors, for example, pCAMBIA, pEAQ-HT, and pBI121, and design these vectors in accordance with the goals of a particular study (Figure 5, Appendix A). Many specialized vectors are intended for specific purposes or plant species (Figure 5, Appendix A). For example, a vector toolkit was developed for monocots; this toolkit comprises the vectors for localization of the protein product utilizing its overexpression, with or without the N-/C-terminal epitope tag (for localizing protein products), and fluorescent fusion expression vectors with EGFP, mVenus, or TagRFP tag (for both C- and N-terminal fusions); the BiFC vectors with (mVN(1–155) and mVC(156–238)) halves; vectors without promoters for analyzing promoter activity as well as native and inducible expression [71]. A recently designed vector toolkit, Plant X-tender, makes it possible to transfer multigene constructs from AssemblX vectors to plant expression vectors and deliver the cassettes carrying several genes to plant cells in a simple and convenient manner [72].

The entire range of the methods for constructing the recombinant DNA molecules has been used to clone target genes into vector constructs, including both classical ones and state-of-the-art techniques for seamless cloning [24,72,73,74,75]. In our view, each of the cloning strategies is applicable to cloning of the target genes into vector constructs, and its selection is, first and foremost, determined by the possibilities and goals of the study [1].

The critical stage in a successful and effective use of transient expression and, in particular, agroinfiltration, is a high expression level of the target sequence (both the target gene and silencing construct). According to the current opinion, a low level of transient expression of heterologous genes frequently results from posttranscriptional gene silencing (PTGS). PTGS is common in plants, being a natural defense mechanism based on RNA interference against viruses and other pathogens. This obstacle can be overcome by the co-expression of PTGS suppressors [24]. Many plant viruses code for the gene products able to inhibit PTGS; however, their mode of action and activity can vary depending on the family of viruses. A powerful suppressor of gene silencing is the p19 protein, coded for by a gene of the *Cymbidium ringspot* virus (*CymRSV*) of tomato, which upregulates the transient expression in various plant species by siRNA sequestration and prevention of their association with the RNA-induced silencing complex (RISC) [50,76].

A comparative study of the viral suppressors of gene silencing in representatives of four different virus families, including Bromoviridae (CMV *2b*), Potyviridae (PRSV HC-Pro), Tombusviridae (TBSV *p19*), and Geminiviridae (TLCV TrAP) was recently performed. Of them, the truncated products of the CMV *2b* gene and TBSV *p19* gene were the most efficient in PTGS suppression; note that both considerably (approximately fourfold) elevate the levels of the reporter protein. The CMV *2b* gene directly interacts with both the RNA and protein components of the silencing pathway [77,78]. Note that the co-infiltration of both truncated products effectively doubles the levels of reporter protein as compared with individual use of either silencing suppressor. This suggests that the combined effect of manifold functions of both proteins—siRNA binding and sequestration, prevention of siRNA duplex assembly into the RISC, and direct interference with the AGO-containing RISC—collectively enhance PTGS suppression [50].

Note that the two main approaches used for the delivery of PTGS suppressors in plant transient gene expression systems utilize either (i) co-transformation with two vectors, one of which carries a silencing suppressor expression cassette and the other, a target gene expression cassette [50] or (ii) expression of the silencing suppressor and target sequence in the same expression cassette [18] (Figure 3 part 3).

According to the current opinion, the efficiency of transient expression depends not only on the components of expression cassette, but also on the following additional conditions: (i) the used agrobacterial strain and the density of its culture for agroinfiltration; (ii) chemical additives in the liquid medium for agroinfiltration; and (iii) plant species and age as well as its growth conditions, including its growth after the agroinfiltration [4,50,79]. Note that although the factors and conditions that provide a more efficient transient expression are mainly assessed using trial and error methods, still some of them are theoretically grounded.

The genetic background of Agrobacterium can significantly influence the ability of the phytopathogen to be the vector for T-DNA and, as a consequence, the efficacy of the first transformation stage (transfection), i.e., the penetration of agrobacteria into the plant cell. A recent study has compared three widespread laboratory strains according to their efficiency in transient expression of a reporter gene in *N. benthamiana*. These strains represent three of the four opine utilizing types, namely, octopine (LBA4404), nopaline (C58C1), and succinamopine (AGL1), and members originated from either of the wild-type progenitor isolates C58 (AGL1 and C58C1) (Figure 6, Appendix A). Of the tested Agrobacterium strains, the hypervirulent AGL1 strain displayed the highest efficiency in transient expression. This suggests that AGL1 is more aggressively disposed for infection or, presumably, has a more efficient bacteria-encoded mechanism for T-DNA transfer as compared with the other two tested strains [50]. Similar studies have been performed for several other plant species [4,80].

The transfection efficiency can be increased by selecting the optimal density of agrobacterial culture and chemical additions to the liquid medium for agroinfiltration. For example, an overdiluted culture may lead to a low bacteria-to-target cells ratio, entailing a decrease in the transformation rate, while concentrated bacterial cultures may induce a bacterial overgrowth and excessive tissue damage [50]. The same model plants were used to demonstrate that acetosyringone, the antioxidant lipoic acid, and surfactant Pluronic F-68 considerably increase the efficiency of target gene transient expression. Although it is unclear how these compounds act, it is assumed that they decrease the surface tension of the cocultivation medium and, possibly, eliminate some substances that inhibit the cell attachment, thereby improving the bacterial invasion and, eventually, T-DNA delivery [50].

According to the available experimental data, the age of a plant and its organs, for example, leaves may considerably influence the transient expression efficiency. As has been convincingly demonstrated, the performance of younger leaves in the transient expression experiments is better; however, since they are, as a rule, of very small size, the first–fourth leaves from the top are usually selected. In addition, the leaves of a flowering plant are not recommended for transient expression experiments because of a lowered transcription and translation levels in the host cells in older leaves and during blossoming, which may have negative effects on the expression efficiency of target genes [79,80,81].

As is recently shown, heat shock of the whole plant (*N. benthamiana*) 1–2 days after agroinfiltration causes a considerable (four–fivefold) increase in the reporter gene expression. Presumably, this positive effect results from the activation of heat shock protein and chaperon expression since these proteins are known players in the plant cell response to extreme temperatures and other abiotic stresses providing the maintenance of homeostasis [82] and enhance a correct folding of native proteins via binding to reactive surfaces of partially folded proteins and effectively sequestering their active sites. This limits the interaction of partially folded intermediates as well as prevents aggregation and degradation of terminally misfolded proteins, which effectively protects them from oxidative stress [50].

Despite a certain success in provision of effective transient expression of target sequences in plants, including selection of the viral gene silencing suppressors, agrobacterial strains, and the conditions for agroinfiltration and plant growth, note that the transient overexpression of a target gene can simply exhaust the host cell transcription and translation machinery, thereby putting at risk the overall expression efficacy. Thus, further studies focused on the development of additional molecular tools able to overcome the negative effects of transient expression are of a paramount importance for technical progress and will enhance a wider use of this research method.

## Figures and Tables

**Figure 1 plants-09-01187-f001:**
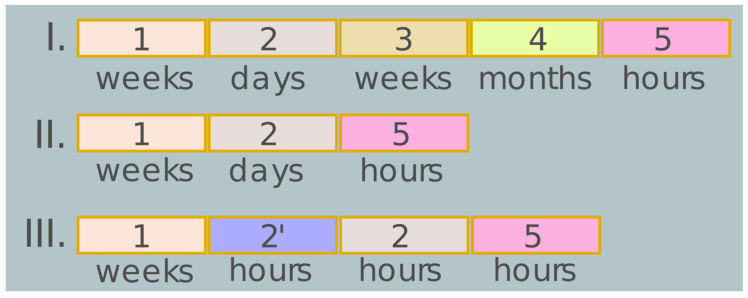
Comparison of the methods for agrobacterial plant transformation (**I**), agroinfiltration (**II**) and protoplast transfection (**III**). Rectangles of the same color and numbers denote the common stages. Time scale is shown below.

**Figure 2 plants-09-01187-f002:**
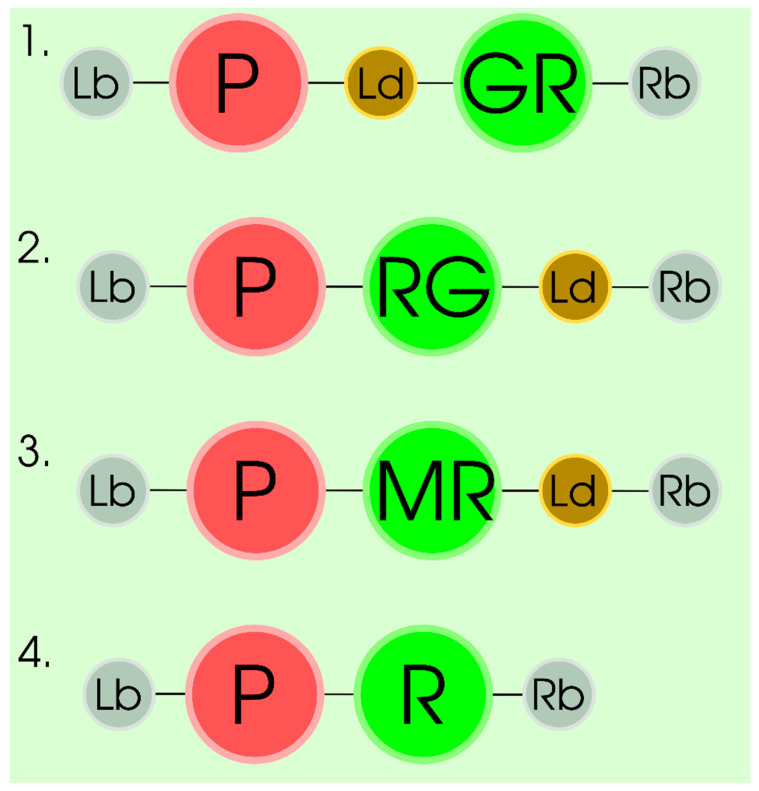
Scheme of T-DNA gene constructs for studying the localization of gene products in a plant cell. 1 and 2—constructs for studying the localization of gene products in a plant cell depending on the localization of the signal peptide of the target gene in the N- or C-terminal region of the target protein, respectively. GR or RG—hybrid gene where the target gene (G) is transcriptionally and translationally fused with the gene coding for reporter protein (R) under the control of a strong constitutive promoter (P), Sp—signal peptide for target protein. 3—positive control, a gene construct carrying the specific markers for plant cell compartments under the control of the same promoter; 4—negative control, a gene construct carrying only the gene coding for a reporter protein under the control of the same promoter; P—a strong constitutive promoter; Lb, R—T-DNA left and right borders.

**Figure 3 plants-09-01187-f003:**
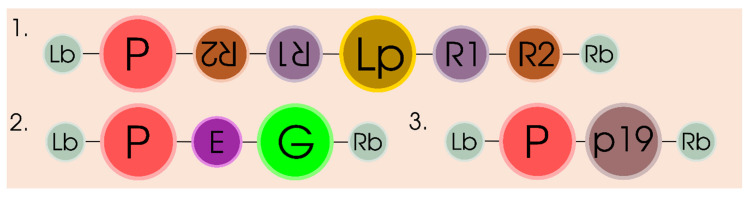
Scheme of T-DNA gene constructs for studying the physiological role of gene products in plants. 1 and 2—constructs for studying the physiological role of gene products in plants due to silencing (1) or overexpression (2) of the target gene, respectively; 3—construct for PTGS suppression, where R1, R2—repeats sequences of target gene forming stem structure; Lp—loop sequence; E—translation enhancer; P—a strong constitutive promoter; G—target gene; p19—gene for PTGS suppression; Lb, Rb—T-DNA left and right borders. Note, construct 3 may represent delivery as co-transformation with two vectors, one of which carries a silencing suppressor expression cassette and the other, a target gene expression cassette [50] or (ii) expression of the silencing suppressor and target sequence in the same expression cassette [24].

**Figure 4 plants-09-01187-f004:**
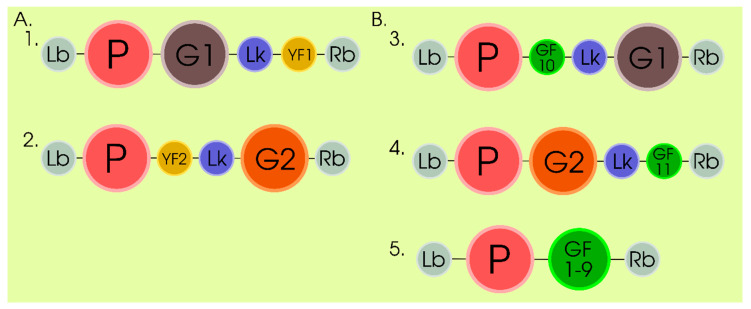
Scheme of T-DNA gene constructs for the study of protein–protein interactions using bimolecular fluorescence complementation (BiFC) analysis (**A**) and tripartite complementation system (**B**). A: 1 and 2—constructs for studying protein–protein interactions in plants using BiFC, where G1, G2—genes of interacting proteins, YF1, YF2— subdomains of YFP (YFP—yellow fluorescent protein). B: 3, 4 and 5 - constructs for studying protein–protein interactions in plants using tripartite system, where G1, G2—genes of interacting proteins, GF1-9, GF10, GF11—subdomains of GFP (GFP – green fluorescent protein). General notation for A and B: P— promoter (for tripartite system—β-estradiol–induced expression, as rule), Lk—flexible linker. Lb, Rb—T-DNA left and right borders. The principle of using systems is described in the text.

**Figure 5 plants-09-01187-f005:**
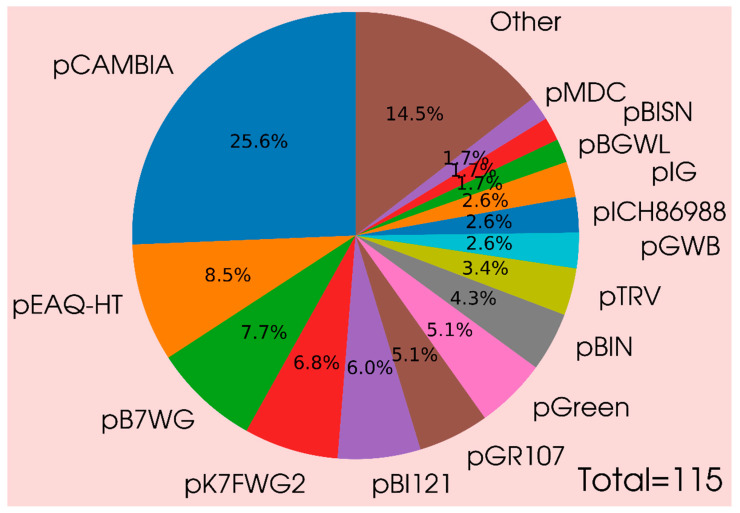
Usage of vectors in transient gene expression in plants. The percentage of vectors usage in transient gene expression in plants is calculated based on scientific articles (Appendix A).

**Figure 6 plants-09-01187-f006:**
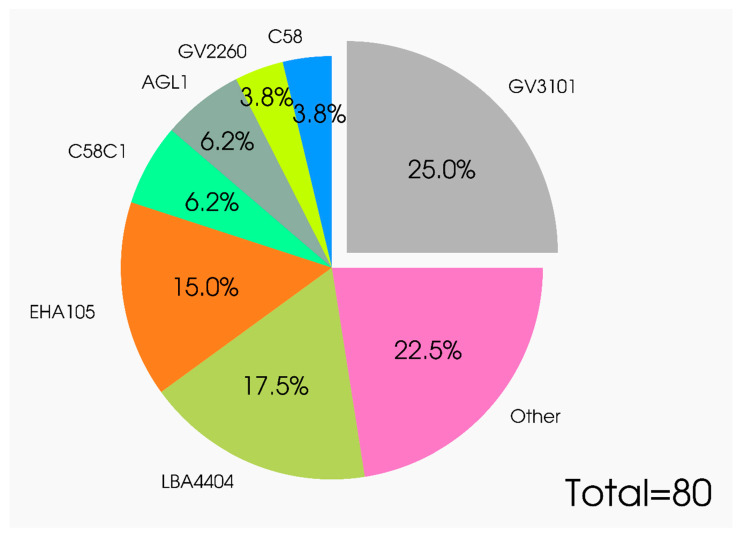
Usage of *A.tumefaciens* strains in transient gene expression in plants. The percentage of agrobacterial strains usage in transient gene expression in plants is calculated based on scientific articles (Appendix A).

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
