# Peer review of "Transient Gene Expression is an Effective Experimental Tool for the Research into the Fine Mechanisms of Plant Gene Function: Advantages, Limitations, and Solutions"

_plants, 2020, doi:10.3390/plants9091187_

Round 1
Reviewer 1 Report
This review highlights the transient gene expression system in plant science research, and gives insights the knowledge, applications, advantages and limitations of this technique into plant gene functions and regulatory pathways. In general, this experimental tool has been used in various areas of plant cell and gene research. The authors have provided a large number of examples to summarize the status of this technique in current molecular biology. However, the information of the transient gene expression system still has not well-organized, and some views and descriptions are incomplete. More latest publications should be included in this review. Specific comments are listed below:
- The author indicated that tools of gene expression manipulation include silent expression and over expression. Beside this, the CRISPR/Cas gene editing technique also has been used in transient and stable -mediated gene knock out expression, such as, rice and wheat. Also, the virus-induced gene silencing (VIGS) with barley stripe mosaic virus (BMSV) is popularly used in transient gene silencing of wheat and barley gene function researches because of the difficulty of stable transformation. Some updated papers should be cited (Fister et al., Front. Plant Sci., 02 March 2018 | https://doi.org/10.3389/fpls.2018.00268; Zhang et al., Nat Commun. 2016 Aug 25;7:12617. doi: 10.1038/ncomms12617; Zhou et al., Nucleic Acids Res. 2014;42(17):10903-14. doi: 10.1093/nar/gku806. And MORE……)
- This manuscript mainly focus on the advantage of this tool in protein localization. Actually, transient gene expression system plays important roles in protein cellular dynamic, trafficking, localization, activity, interaction, regulation and biological functions. For example, the plant pollen tube is also a good model for protein localization, trafficking and plant cell growth in terms of transient gene expression, which is generally used in Arabidopsis, Tobacco and Rice (Wang and Jiang, Nat Protoc. 2011 doi:10.1038/nprot.2011.309. Li et al. Plant Physiol 2018, DOI: https://doi.org/10.1104/pp.17.01759). Another examples, one latest publication showed that the protoplast also is a good model for ‘gene-for-gene’ research in plant immunity (Saur, et al., 2019, Plant METHODS, https://doi.org/10.1186/s13007-019-0502-0). Thus, the current summary of advantages in transient gene expression system are vague and inadequate.
- Fluorescence tag and report genes. The authors only discussed the GFP/YFP/mCherry. Beside these, VENUS, CFP, GUS and Luciferase (Dual-LUC system) are also used in plant protein activity, gene promoter activity, target gene expression regulation, protein-protein interactions, and effector-reporter recognitions (huge publications). This manuscript is partially summarized these report genes. The investigation of these tags/report genes in current publications and clear classification of transient gene expression system in plant molecular research have a high significance, rather than the current information in this manuscript, such as, species, vector type (Fig 5), agrobacterial couture (Fig 6).
- Line 65-71 ‘However, despite an apparent simplicity of 65 production of transgenic plants, a wide use of this technology has certain main limitations: (i) the 66 dependence of transformation efficiency on plant genotype and (ii) considerable material and time 67 expenditures when constructing plant transgenic lines, including selection of true transgenic 68 individuals among a tremendous number of primary transformants, confirmation of integration of 69 the transferred sequences and of the events of integration of a single copy into the plant genome, 70 determination of the expression level for both mRNA and protein product, and so on (Figure. 1, I)’. Even the stable transformation has some limitations, transgenic plants are required for gene function validation, phenotyping and genetic, which can not be replaced by transient gene expression. Thus, the author could compare this two techniques, such as time, methods, but both of them are important for plant science research.
- Line 133-136 ‘Currently, it would be unjust to speak about insufficient number of 113 publications describing similar basic protocols of transient expression for different plant species as 114 well as the technical aspects associated with the effective use of this approach in assessing the 115 functional role of the gene(s) in individual plant species [3-5].’ I do not agree with this statement. The listed references did not include the model plants, like Arabidopsis, tobacco, onion, rice etc. There are large publications of these research (Zhang et al, 2020, Plant Comms, https://doi.org/10.1016/j.xplc.2020.100028).
- All gene name should be ITALIC. The author should check this point throughout the text.
- Why some non-English characters exist in Figure legend? Like 250, 252, 254, Please check throughout the text.
Author Response
Please, see the attachment

Reviewer 2 Report
Please check the english language.
Also, a lot of sentences are very long. The content is ok, but authors need to shorten some long parts in the manuscript.
The quality and clarity of figures have to be enhanced.
Author Response
Please, see the attachment
Reviewer 3 Report
The manuscript „Transient gene expression is an effective experimental tool for the research into the fine mechanisms of plant gene function: advantages, limitations, and solutions” is based on an extensive literature review citing more than 400 studies conducted within this research field. The authors provide an overview of the involved technology comprising the gene constructs, delivery systems and host plants as well as applications of transient transformation for basic research to elucidate gene function, subcellular protein localization, and protein-protein interactions.
The manuscript is well structured and illustrated with informative figures. The supplementary tables provide a comprehensive overview of the current state of the art. I believe this manuscript is a timely review on this important technology and will therefore be of interest for a wide audience. My only advice would be to obtain language editing from a native speaker to improve the readability of the manuscript.
Minor points:
- Figure 1 I: the labeling of the fourth box should read “months”
- Figure 2: the caption contains words in Cyrillic script: please translate; the term “leading sequence” should be replaced by “signal peptide” as the more appropriate designation. This also applies to other sections in the manuscript beside this figure legend.
- Figure 3: please translate Cyrillic script
- Figure 4: please translate Cyrillic script
- Figure 5 and 6: the figure legends seem to be interchanged. Please double check.
Round 2
Reviewer 1 Report
I only have a few minor comments.
1. line 117-118, ‘other important aspects of plant molecular biology’, What are other important aspects? Please specify them.
2. Line 242-246, GFP only acts the role of tag or reporter, how to be effective marker? It is not accurate that YFP and mCherry are the most suitable for performing localization in peroxisomes. The cited papers did not compare the signal efficiency between GFP and YFP/mCherry. There are also most papers to investigate the peroxisomes protein using GFP.
3.Line 251, 310, 486, 487, 489, what is ‘и’? non-English characters.
